# Establishing Quality Assurance for HIV-1 Rapid Test for Recent Infection in Thailand through the Utilization of Dried Tube Specimens

**DOI:** 10.3390/diagnostics14121220

**Published:** 2024-06-09

**Authors:** Supaporn Suparak, Petai Unpol, Kanokwan Ngueanchanthong, Sirilada Pimpa Chisholm, Siriphailin Jomjunyoung, Wipawee Thanyacharern, Nitis Smanthong, Pojaporn Pinrod, Kriengkrai Srithanaviboonchai, Thitipong Yingyong, Theerawit Tasaneeyapan, Somboon Nookhai, Archawin Rojanawiwat, Sanny Northbrook

**Affiliations:** 1Department of Medical Sciences, Ministry of Public Health, Nonthaburi 11000, Thailand; petai.u@dmsc.mail.go.th (P.U.); kanokwan.n@dmsc.mail.go.th (K.N.); sirilada.p@dmsc.mail.go.th (S.P.C.); wipawee.t@dmsc.mail.go.th (W.T.); pojaporn.p@dmsc.mail.go.th (P.P.);; 2Research Institute for Health Sciences, Chiang Mai University, Chiang Mai 50200, Thailand; ksrithanaviboonchai@gmail.com; 3Department of Disease Control, Ministry of Public Health, Nonthaburi 11000, Thailand; 4Division of Global HiV & Tuberculosis, U.S. Centers for Disease Control and Prevention, Nonthaburi 11000, Thailandhpz3@cdc.gov (S.N.);

**Keywords:** dried tube specimen, recent infection, quality control, external quality assessment

## Abstract

The present study focuses on establishing the quality assurance of laboratories for recent infections (RTRI) in Thailand. We developed a cold-chain independent method, using fully characterized plasma obtained from the Thai Red Cross Society, and prepared as dried tube specimens (DTS). Twenty microliters of HIV-seronegative, recent, and long-term infected samples were aliquoted into individual tubes and dried at room temperature, 20–30 degrees Celsius, in a biosafety cabinet overnight to ensure optimal preservation. The DTS external quality control and external quality assessment were tested for homogeneity and stability following the ISO/Guide 35 guidelines. The DTS panels were distributed to 48 sites (FY 2022) and 27 sites (FY 2023) across 14 and 9 provinces, respectively, in Thailand. The results from participating laboratories were collected and evaluated for performance. The results were scored, and acceptable performance criteria were defined as the proportion of panels correctly tested, which was set at 100%. The satisfactory performance ranged from 96% to 100% and was not significantly different among the 13 health regions. The developed and implemented DTS panels can be used to monitor the quality of RTRI testing in Thailand.

## 1. Introduction

Despite being a leader in the region in responding to and adopting several policies and measures to accelerate HIV epidemic control, Thailand is estimated to have approximately 562,000 people living with HIV, with 9200 new infections in 2022 [1,2]. Continuous assessment of a population’s HIV epidemic through ongoing surveillance will remain essential to ensure that interventions are efficiently and effectively targeting those at the highest risk of acquiring or transmitting HIV. Integrating strategies to locate hard-to-reach groups and newly infected HIV cases, and rapidly intervening to stop the chain of transmission, can help to control the spread of HIV.

The Rapid Test for Recent Infection (RTRI) for HIV can be used to detect individuals who have been recently infected and can be employed to identify HIV transmission hot spots and help to target HIV prevention and treatment resources [3]. The test demonstrates a sensitivity of 99.1% (95% CI: 98.0–99.6%) and has a specificity of 98.9% (95% CI: 98.1–99.4%) for HIV diagnosis [3]. Moreover, it has shown excellent performance in routine HIV testing services, specifically for the real-time surveillance of recent HIV infections among pregnant adolescent girls and young women attending antenatal clinics in Malawi [4].

The RTRI is a single device that can be used at the point of care. While the test kit incorporates a built-in procedural control in the strip device to establish assay validity, its use remains necessary to monitor and enhance clinical laboratory practices. The objectives of this research are to develop the external Quality Control (QC) and External Quality Assessment (EQA), a cold-chain independent method using plasma prepared as dried tube specimens (DTS), and to implement it to ensure quality assurance in RTRI testing in Thailand.

## 2. Materials and Methods

### 2.1. HIV-1 Test Samples

HIV-seropositive and seronegative plasma samples, preserved in acid citrate dextrose anticoagulant, were obtained from de-identified blood units that had been discarded by blood donors. These samples were collected by the Thai Red Cross Society. Although we utilized de-identified blood samples from blood donors, we acknowledge that our research involves human subjects. Our study was conducted in compliance with ethical standards and was approved by the Research Ethics Committee of the Department of Medical Sciences, Ministry of Public Health (Study code 12/2565).

The plasma specimens underwent testing in accordance with the Thai National Guidelines, utilizing the following tests: Elecsys HIV Combi PT (Roche Diagnostics Ltd., Mannheim, Germany), followed by Serodia HIV-1/2 MIX (Fujirebio Inc., Tokyo, Japan), and Alere HIV Combo (Alere Medical Co., Ltd., Chiba, Japan), following the instructions provided in the respective packages at the Transfusion-Transmitted-Pathogens-Section, Department of Medical Sciences, Ministry of Public Health, Thailand. The concordance sample results of HIV-seropositive and negative specimens were then tested using the Asanté™ HIV-1 Rapid Recency^®^ Assay (Sedia Biosciences Corporation, Beaverton, OR, USA) to classify them as HIV-1 recent, HIV long-term, or HIV-negative. The protocol of this assay followed the instructions in the Asanté™ HIV-1 Rapid Recency^®^ Assay package insert [5].

The Asanté Rapid Recency assay is formatted as a lateral flow device with three lines, representing a Control line (C), a Positive Verification line (V), and a Long-Term line (LT/R) to distinguish recent from long-term infection. The limitation of antigen amount applied to the LT/R line serves as the basis for distinguishing recent (low-avidity antibodies) from long-term (high-avidity antibodies) infection. Interpretation of the Asanté™ HIV-1 Rapid Recency^®^ Assay results is based on the presence or absence of specific lines: the control line (C Line), the positive verification line (V Line), and the long-term/recent line (LT/R), respectively, by visual reading (Figure 1d).

Recent HIV infection is defined as the period of ≤6 months after infection; beyond this timeframe, it is considered long-term infection [3]. Since we were unable to obtain information from de-identified blood samples from blood donors, the laboratory testing results were used as the reference for diagnosis and classification.

The HIV-1 RTRI test kit is designed to identify both HIV-1 and HIV-2 infections. However, it should be noted that the test may not be able to identify recent infections for HIV-2. Fortunately, there have been no reported incidences of HIV-2 infection in Thailand; thus, there are no concerns about false recent results caused by HIV-2 infection in our study population.

### 2.2. Preparation of HIV Dried Tube Specimens (DTS) QC and EQA Panel

The identified HIV-recent, HIV-long-term, and HIV negative samples were prepared as dried tube specimens (DTS) and defined for QC and EQA panels.

The QC was defined as utilizing known result panels to evaluate the performance of the test kit. These known result panels were used to monitor the precision and accuracy of the laboratory testing procedures by comparing the obtained results with the expected values.

On the other hand, the EQA involved the use of unknown result panels provided by external quality assessment schemes. These panels were distributed to laboratories without prior knowledge of their results. Through EQA, we assessed not only the accuracy and reliability of the testing procedures but also the overall competency of our laboratory personnel in handling and analyzing samples.

In total, 18 QC samples classified by sample types (6 each for negative, recent, and long-term categories) for 6 rounds, and 18 EQA samples for 5 rounds, were defined. These samples were meticulously selected to ensure representative coverage of the various categories and rounds of assessment.

For the DTS preparation, only the plasma samples confirmed as HIV-recent and long-term positive were subjected to inactivation at 56 °C for 30 min, while the HIV seronegative samples remained untreated. Preparation of the DTS was modified from that used by Parekh et al. [6] Green food coloring was added to the plasma samples at a concentration of 0.001% for easy visual identification. Twenty microliters of each sample were aliquoted into individual tubes, then dried at room temperature in a biosafety cabinet overnight or until dry. The DTS were reconstituted with 200 µL of PBS-0.05% Tween before testing for HIV serostatus.

Three samples each of DTS HIV-recent, long-term, and negative samples were subjected to testing in triplicate, and the results were compared with plasma liquid samples using Elecsys HIV Combi PT (Roche Diagnostics Ltd., Mannheim, Germany), Alere HIV Combo (Alere Medical Co., LTD., Chiba, Japan), and the Asanté™ HIV-1 Rapid Recency^®^ Assay. These tests were conducted following the instructions provided by the respective manufacturers.

### 2.3. Homogeneity and Stability of HIV DTS QC and EQA Panel

To ensure the consistency of results in all HIV DTS tubes for RTRI, a homogeneity test was performed according to ISO/Guide 35. This involved sampling HIV DTS negative, recent, and long-term samples using the ∛*n* formula [7]. Ten randomized samples were tested with the Asante HIV-1 Rapid Recency Assay, and all results were checked for concordance. Stability testing of the HIV DTS for RTRI was conducted. The samples were stored at temperatures of −20 °C, 2–8 °C, 20–30 °C, 37 °C, and 45 °C for eight weeks, and HIV recent infection testing was conducted using the Asanté™ HIV-1 Rapid Recency^®^ Assay on a weekly basis.

### 2.4. Implementation of HIV DTS QC and EQA

Laboratory site selection for RTRI testing was voluntary and focused in areas with high HIV burdens in the country. In FY2022, 48 laboratories participated across 14 provinces, covering all 13 health regions. However, in FY2023, due to budget optimization and the reported number of new HIV cases, only 27 RTRI lab hubs were selected in 9 provinces. The DTS QC and EQA programs were implemented in RTRI lab hubs to monitor the accuracy of the HIV-1 RTRI tests.

### 2.5. Packaging and Shipping of HIV DTS QC and EQA Panel

The QC panels were dispatched to laboratories at ambient temperature over 6 rounds from Apr 2022 to Apr 2023, each consisting of 3 QC samples: QC negative, QC recent, and QC long-term samples. This distribution resulted in a total of 18 QC samples, categorized by sample types, with 6 samples allocated for each category (negative, recent, and long-term).

The EQA panels comprised 18 samples distributed across 5 rounds. Specifically, there were 4 samples each in A-2022 (Apr 2022), B-2022 (Jun 2022), and C-2022 (Aug 2022), and 3 samples each in A-2023 (Jun 2023), and B-2023 (July 2023).

Regarding the discrepancy in the number of DTS tubes distributed across different years, it is important to note that this variation is due to differences in the distribution schedule rather than the number of tests conducted. The quality-assurance assessment results were presented on a laboratory basis, not based on the number of tests. Therefore, the variation in the number of DTS tubes distributed across different years did not affect the assessment of laboratory performance.

The DTS samples panel was packaged along with tube reconstitution buffer (phosphate-buffered saline with Tween 20-PBS-0.05% Tween) and a testing instruction sheet. The DTS panels were shipped at room temperature to the sites. For the DTS QC/EQA procedure, the DTS were reconstituted by adding 200 μL of PBS-0.05% Tween to each specimen tube, mixed by vortex, and the DTS pellets were allowed to dissolve at room temperature for 5–10 min. The reconstituted DTS were then tested with the Asanté™ HIV-1 Rapid Recency^®^ Assay.

### 2.6. Evaluating the Laboratory Performance

A site performance score achieving 100 points was considered to be a satisfactory performance. The performance rate was defined as the proportion of panels correctly tested, both in visual band reading and in the interpretation of results reported on time before the closing date. The satisfactory results were expressed as a percentage of the individual site scores. An investigation into the cause of satisfactory performances below 100% was conducted for thorough monitoring.

### 2.7. Statistical Analysis

Descriptive analyses of testing sites by site type, participation and performance rates were conducted for each round. All statistical analyses used two-tailed tests, with *p* ≤ 0.05 considered as statistically significant. Data analysis was conducted using IBM SPSS Statistics version 21.0 (IBM Corporation, Armonk, NY, USA).

## 3. Results

### 3.1. Characterization of DTS QC/EQA for HIV-1 RTRI

The plasma specimens underwent testing with Elecsys HIV Combi PT (Roche Diagnostics Ltd., Mannheim, Germany), followed by Serodia HIV-1/2 MIX (Fujirebio Inc., Japan), and Alere HIV Combo (Alere Medical Co., Ltd., Chiba, Japan) to identify HIV-seropositive and negative specimens. The results demonstrated concordance in HIV-seropositive and negative status across all three tests. HIV-seropositive samples were further classified as recent or long-term using the Asanté™ HIV-1 Rapid Recency^®^ Assay. The identified recent, long-term, and negative samples were prepared as dried tube specimens (DTS).

When comparing the developed DTS with the liquid plasma results, both samples were tested for anti-HIV antibodies using the Elecsys HIV Combi PT and Alere HIV Combo tests, as well as the Asanté™ HIV-1 Rapid Recency^®^ Assay. The results indicated that the drying process did not alter the HIV serostatus result, demonstrating concordant results between the liquid and DTS samples (Table 1).

### 3.2. Homogeneity and Stability of HIV DTS QC/EQA for HIV-1 RTRI

The homogeneity results from ten tubes each of HIV-negative, HIV-positive recent, and HIV-positive long-term samples showed consistency, suggesting uniformity in the prepared DTS samples (Table 2).

The stability of DTS QC/EQA was assessed using the Asanté™ HIV-1 Rapid Recency^®^ Assay. Results revealed that the negative sample remained stable at all temperatures for 8 weeks, while the recent samples demonstrated stability for 8 weeks at −20 °C, 2–8 °C, 20–30 °C, 37 °C, and for 2 weeks at 45 °C. Long-term samples exhibited stability for 6 weeks at −20 °C, 2–8 °C, 20–30 °C, 37 °C, and for 2 weeks at 45 °C (Table 3).

The results of the 6-week stability test are depicted in Figure 1. Based on the stability study, it was determined that the samples could be sent to the sites at ambient temperatures (RT, 20–30 °C) for 7 days without affecting the test results.

### 3.3. QC HIV-1 RTRI Performance

QC samples with known results allow laboratories to monitor the accuracy and precision of their testing procedures by comparing obtained results with expected values.

The QC HIV-1 RTRI results were collected and evaluated from 48 participating laboratories in FY22 and 27 in FY23 across 13 Health Regions in six rounds (Table 4). The criteria for achieving 100% satisfactory results included the proportion of panels correctly tested, both in visual band reading and interpretation of results, as well as reporting on time before the closing date. The overall satisfactory performance of RTRI testing labs for each round ranged from 96% to 100%, with no significant differences observed across the six rounds (*p* > 0.05). Although the site performance rates improved over time, they were inconsistent from round to round and varied between health regions.

For the total of 18 QC samples classified by sample types (6 each for negative, recent, and long-term categories, see Table 5), performance varied across health regions and laboratories, resulting in discordant results in rounds A-2022, B-2022, and C-2022. In round A-2022, labs from regions 7 and 13 reported incorrect results for QC negative samples. In round B-2022, a lab from region 7 reported an incorrect result for the QC long-term sample. In round C-2022, a lab from region 7 reported an incorrect result for the QC negative sample, and a lab from region 13 reported incorrect results for both the QC recent and long-term samples. QC samples help to identify errors or deviations in testing procedures promptly, allowing for corrective actions to be taken to maintain the quality of results.

### 3.4. EQA HIV-1 RTRI Performance

EQA samples provide an external benchmark with which to assess the accuracy and reliability of testing procedures across different laboratories. EQA samples provide an independent assessment of laboratory performance, helping to identify systemic issues or areas for improvement.

EQA panels were distributed for 5 rounds with 4 samples in A-2022 (Apr 2022), B-2022 (Jun 2022), C-2022 (Aug 2022), and 3 samples in A-2023 (Jun 2023), B-2023 (July 2023) to labs across 13 health regions.

The criteria for achieving 100% satisfactory results involve the proportion of correctly tested panels. Average performance rates among the health regions’ sites ranged from 96% to 100% (Table 6) and were consistent across the five rounds (*p* > 0.05).

A detailed breakdown of individual lab performances across the 13 health regions demonstrates the observed variations. The following notable fluctuations from round to round: in A-2022, Health Region 13’s lab achieved 93%. In B-2022, Health Region 5’s lab showed 75% performance. Similarly, in C-2022, Health Region 12’s lab demonstrated a 75% performance. In B-2023, Health Region 11’s lab exhibited a 75% performance.

For the total of 18 EQA samples, performance varied across rounds, and discordant results were found in all sample types (Table 7). The labs that did not meet the criteria were closely monitored and advised to adhere to the protocol instructions for HIV-1 RTRI testing. Primary causes of errors included non-compliance with HIV-1 RTRI testing instructions and inadequate dissolution of the DTS sample panel before testing. Additionally, procedural errors occurred when the test strip was dipped into the sample tube without the recommended buffer as per the instructions. Some labs also demonstrated incorrect interpretation. EQA programs assess not only the accuracy of testing procedures but also the competency of laboratory personnel in handling and analyzing samples. Participation in EQA helps to ensure that laboratory staff are proficient in their roles and maintain high standards of performance.

## 4. Discussion

Quality assurance is the implementation of planned activities to support the quality-management system [8,9]. EQA and quality control are integral components of quality assurance, ensuring the accuracy of tests and detecting errors. The quality of laboratory testing is paramount, with direct implications for prevention, treatment, and transmission to new individuals, especially in point-of-care testing [10]. It is highly recommended that a point-of-care (POC) quality-assurance plan addresses the entire quality-assurance management cycle, including external quality assurance as a relevant activity in POC testing. The results serve as a strong indicator of laboratory quality [11].

POC devices have built-in internal QC to monitor the performance of an instrument in real time and, to some extent, the user. Internal quality controls for HIV-1 RTRI POC testing rely on the control line, but the challenge across sites lies in the skill required for visual interpretation [3,4]. A successful quality-control program hinges on stable, optimized QC samples and addresses factors that may limit the quality assessment of RTRI testing [9,12,13,14]. External QC serves as a measure of imprecision or unexpected variation in the testing process, and any sources of variation are investigated [15,16,17]. In the present study, we developed a tailored DTS-based external QC and EQA program for recent HIV infection, aiming for its deployment in point-of-care settings to enhance diagnostic quality. The EQA scheme monitors laboratory quality by sending blinded, known-status samples to participating laboratories and analyzing the returned results [18]. EQA challenges the testing environment, examines the processes, and provides remediation [19,20]. Efficient EQA and quality-control systems assure public health officials that HIV diagnostic tools provide reliable results, instilling confidence in the outcomes [9,12,21].

The external QC and EQA for HIV-1 RTRI were implemented to monitor the quality of HIV recency testing for the surveillance of recent HIV infections in Thailand. We found that performance rates varied among the labs and health regions throughout the study period. A low score alerted the laboratory to identify the problem, and corrective action was then taken. Errors arose from testing procedures or from incorrect data entry, notably in the April 2022 round. Immediate investigation and remedial action were taken to correct the problem before patients’ results were adversely affected. When errors were identified in testing procedures, laboratories were advised to reassess sample-testing processes, including sample reconstitution and RTRI testing. Following the implementation of these recommendations, there were no instances of recurrent errors reported by the same laboratories, indicating an enhancement in the quality of the testing.

Regular participation in QC and EQA programs fosters a culture of continuous improvement within laboratories. Feedback from QC and EQA results can be used to refine testing protocols, address training needs, and implement best practices to enhance overall laboratory performance.

## 5. Conclusions

We established the first implementation of external quality-control and external quality assessment for the national rollout of an HIV recent infection surveillance program using DTS. The combined use of QC and EQA programs strengthens the quality-management systems of laboratories, enhances the reliability of test results, and ultimately improves patient care and outcomes. The developed DTS sample panel was conveniently transported at room temperature and did not require regulation as infectious substances for shipping. An ongoing review of laboratory performance will ensure that the identified problems are addressed to prevent consistent issues from recurring.

These programs can be particularly useful in resource-limited settings. As the country strives to achieve the UNAIDS targets by 2030, it is crucial to adopt and rapidly scale up comprehensive quality programs for the HIV rapid recent infection testing continuous quality improvement initiative.

## Figures and Tables

**Figure 1 diagnostics-14-01220-f001:**
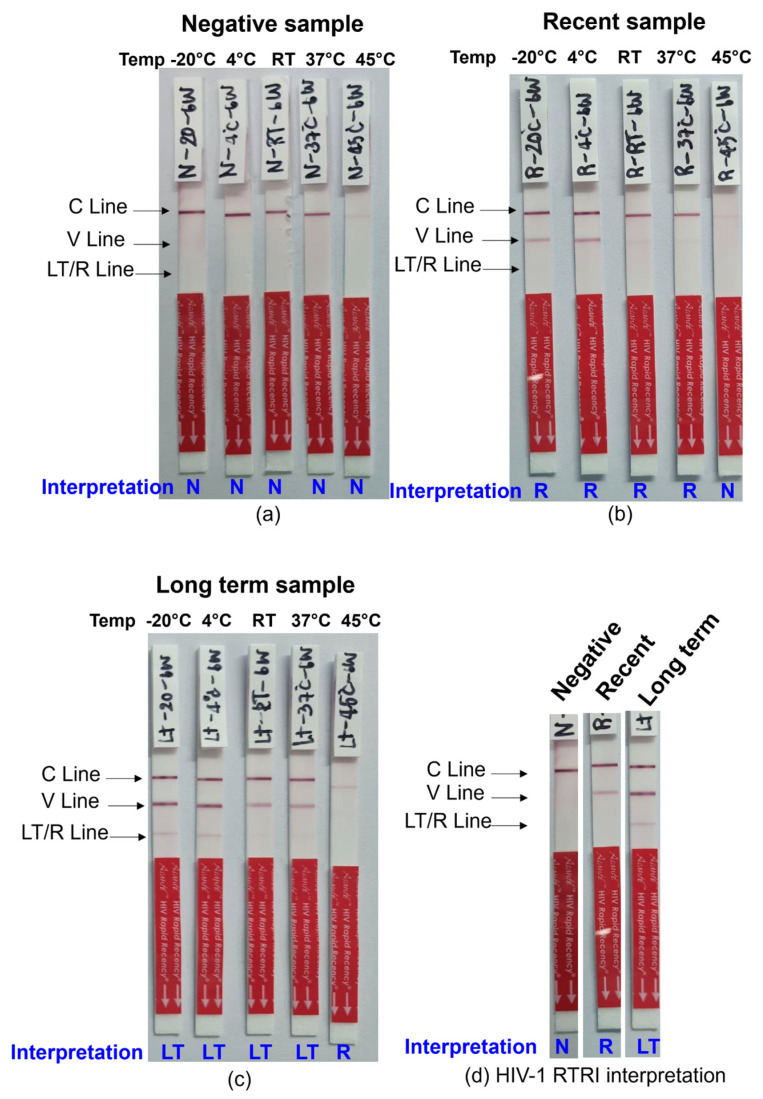
Stability of DTS QC/EQA for HIV-1 RTRI was assessed using the Asanté™ HIV-1 Rapid Recency^®^ Assay. The samples, including: (**a**) negative; (**b**) recent, and (**c**) long-term, demonstrated stability at 2–8 °C, 30 °C, and 37 °C for a minimum of 6 weeks; (**d**) the interpretation of the Asanté™ HIV-1 Rapid Recency^®^ Assay results rely on the visual detection of specific lines: the control line (C Line), the positive verification line (V Line), and the long-term/recent line (LT/R). As a result, the interpretation of results were confirmed by three independent individuals.

**Table 1 diagnostics-14-01220-t001:** Characterization results of HIV DTS for RTRI.

RTRI Panel	Sample Type	Anti-HIV Test Results		HIV-1 RTRI Results			
		Elecsys HIV Combi PT	Alere HIV Combo	Asanté™ HIV-1 Rapid Recency^®^ Assay			
		Interpretation *	Interpretation	C	V	LT/R	Interpretation
Recent Sample (N = 3)	Liquid	Reactive	Reactive	+	+	−	Recent
	DTS	Reactive	Reactive	+	+	−	Recent
Long-term Sample (N = 3)	Liquid	Reactive	Reactive	+	+	+	Long-term
	DTS	Reactive	Reactive	+	+	+	Long-term
Negative (N = 3)	Liquid	Non-reactive	Non-reactive	+	−	−	Negative
	DTS	Non-reactive	Non-reactive	+	−	−	Negative

* Reactive: COI ≥ 1.0, C = control line, V = verification line, LT = Long term, R = recent. + = Presence of specific lines, − = Absence of specific lines.

**Table 2 diagnostics-14-01220-t002:** Homogeneity results of DTS QC/EQA for HIV-1 RTRI, Thailand.

Tube No.	HIV Negative Samples				HIV-1 Recent Samples				HIV-1 Long-Term Samples			
	C	V	LT/R	Interpretation	C	V	LT/R	Interpretation	C	V	LT/R	Interpretation
1	+	−	−	Negative	+	+	−	Recent	+	+	+	Long-term
2	+	−	−	Negative	+	+	−	Recent	+	+	+	Long-term
3	+	−	−	Negative	+	+	−	Recent	+	+	+	Long-term
4	+	−	−	Negative	+	+	−	Recent	+	+	+	Long-term
5	+	−	−	Negative	+	+	−	Recent	+	+	+	Long-term
6	+	−	−	Negative	+	+	−	Recent	+	+	+	Long-term
7	+	−	−	Negative	+	+	−	Recent	+	+	+	Long-term
8	+	−	−	Negative	+	+	−	Recent	+	+	+	Long-term
9	+	−	−	Negative	+	+	−	Recent	+	+	+	Long-term
10	+	−	−	Negative	+	+	−	Recent	+	+	+	Long-term

C = control line, V = verification line, LT = Long term, R = recent. + = Presence of specific lines, − = Absence of specific lines.

**Table 3 diagnostics-14-01220-t003:** Stability result of DTS QC/EQA for HIV-1 RTRI at different temperatures, Thailand.

RTRI Panel	Temp	−20 °C	2–8 °C	Room Temp	37 °C	45 °C
Negative	Week 2	Negative	Negative	Negative	Negative	Negative
	Week 4	Negative	Negative	Negative	Negative	Negative
	Week 6	Negative	Negative	Negative	Negative	Negative
	Week 8	Negative	Negative	Negative	Negative	Negative
Recent	Week 2	Recent	Recent	Recent	Recent	Recent
	Week 4	Recent	Recent	Recent	Recent	* Negative
	Week 6	Recent	Recent	Recent	Recent	* Negative
	Week 8	Recent	Recent	Recent	Recent	* Negative
Long-term	Week 2	Long-term	Long-term	Long-term	Long-term	Long-term
	Week 4	Long-term	Long-term	Long-term	Long-term	* Recent
	Week 6	Long-term	Long-term	Long-term	Long-term	* Recent
	Week 8	Long-term	Long-term	Long-term	* Recent	* Negative

* The samples exhibited instability.

**Table 4 diagnostics-14-01220-t004:** QC HIV-1 RTRI performance in 13 Health regions, Thailand.

Health Regions	A-2022 (Apr 2022)	B-2022 (May 2022)	C-2022 (Jun 2022)	D-2022 (July 2022)	E-2022 (Aug 2022)	A-2023 (Apr 2023)
	No. Lab Passed/No. Lab Total (%)					
Total	46/48 (96)	47/48 (98)	46/48 (96)	48/48 (100)	46/46 (100)	27/27 (100)
Health region 1	3/3 (100)	2/3 (67)	3/3 (100)	3/3 (100)	3/3 (100)	2/2 (100)
Health region 2	2/2 (100)	2/2 (100)	2/2 (100)	2/2 (100)	1/1 (100)	NA *
Health region 3	3/3 (100)	3/3 (100)	3/3 (100)	3/3 (100)	3/3 (100)	3/3 (100)
Health region 4	2/2 (100)	2/2 (100)	2/2 (100)	2/2 (100)	2/2 (100)	1/1 (100)
Health region 5	4/4 (100)	4/4 (100)	4/4 (100)	4/4 (100)	4/4 (100)	NA *
Health region 6	1/1 (100)	1/1 (100)	1/1 (100)	1/1 (100)	1/1 (100)	1/1 (100)
Health region 7	2/3 (67)	3/3 (100)	2/3 (67)	3/3 (100)	2/2 (100)	2/2 (100)
Health region 8	1/1 (100)	1/1 (100)	1/1 (100)	1/1 (100)	1/1 (100)	1/1 (100)
Health region 9	2/2 (100)	2/2 (100)	2/2 (100)	2/2 (100)	2/2 (100)	NA *
Health region 10	1/1 (100)	1/1 (100)	1/1 (100)	1/1 (100)	1/1 (100)	NA *
Health region 11	7/7 (100)	7/7 (100)	7/7 (100)	7/7 (100)	7/7 (100)	4/4 (100)
Health region 12	4/4 (100)	4/4 (100)	4/4 (100)	4/4 (100)	4/4 (100)	2/2 (100)
Health region 13	14/15 (93)	15/15 (100)	14/15 (93)	15/15 (100)	15/15 (100)	11/11 (100)

* NA: Not analyzed.

**Table 5 diagnostics-14-01220-t005:** QC HIV-1 RTRI results at 48 sites in FY22 and 27 sites in FY23, Thailand.

RTRI Panel Sample	A-2022 (Apr 2022)		B-2022 (May 2022)		C-2022 (Jun 2022)		D-2022 (July 2022)		E-2022 (Aug 2022)		A-2023 (Apr 2023)	
	No. Lab Passed/No. Lab Total	%	No. Lab Passed/No. Lab Total	%	No. Lab Passed/No. Lab Total	%	No. Lab Passed/No. Lab Total	%	No. Lab Passed/No. Lab Total	%	No. Lab Passed/No. Lab Total	%
Negative	46/48	96	48/48	100	47/48	98	48/48	100	46/46	100	27/27	100
Recent	48/48	100	48/48	100	47/48	98	48/48	100	46/46	100	27/27	100
Long term	48/48	100	47/48	98	47/48	98	48/48	100	46/46	100	27/27	100

**Table 6 diagnostics-14-01220-t006:** EQA HIV-1 RTRI performance in 13 health regions, Thailand.

Health Region	A-2022 (Apr 2022)	B-2022 (May 2022)	C-2022 (Jun 2022)	A-2023 (Jun 2023)	B-2023 (July 2023)
	No. Lab Passed/No. Lab Total (%)				
Total	47/48 (98)	47/48 (98)	46/48 (96)	27/27 (100)	26/27 (96)
Health region 1	3/3 (100)	3/3 (100)	3/3 (100)	2/2 (100)	2/2 (100)
Health region 2	2/2 (100)	2/2 (100)	2/2 (100)	NA *	NA *
Health region 3	3/3 (100)	3/3 (100)	3/3 (100)	3/3 (100)	3/3 (100)
Health region 4	2/2 (100)	2/2 (100)	2/2 (100)	1/1 (100)	1/1 (100)
Health region 5	4/4 (100)	3/4 (75)	4/4 (100)	NA *	NA *
Health region 6	1/1 (100)	1/1 (100)	1/1 (100)	1/1 (100)	1/1 (100)
Health region 7	3/3 (100)	3/3 (100)	3/3 (100)	2/2 (100)	2/2 (100)
Health region 8	1/1 (100)	1/1 (100)	1/1 (100)	1/1 (100)	1/1 (100)
Health region 9	2/2 (100)	2/2 (100)	2/2 (100)	NA *	NA *
Health region 10	1/1 (100)	1/1 (100)	1/1 (100)	NA *	NA *
Health region 11	7/7 (100)	7/7 (100)	7/7 (100)	4/4 (100)	3/4 (75)
Health region 12	4/4 (100)	4/4 (100)	3/4 (75)	2/2 (100)	2/2 (100)
Health region 13	14/15 (93)	15/15 (100)	14/15 (93)	11/11 (100)	11/11 (100)

* NA: Not analyzed; The laboratory chose not to participate.

**Table 7 diagnostics-14-01220-t007:** EQA HIV-1 RTRI performance in each panel.

Sample ID	HIV-1 Test Samples	No. Lab Passed/ No. Lab Total	%	Sample ID	HIV-1 Test Samples	No. Lab Passed/ No. Lab Total	%	Sample ID	HIV-1 Test Samples	No. Lab Passed/ No. Lab Total	%
A-2022 (Apr 2022)				B-2022 (Jun 2022)				C-2022 (Aug 2022)			
R651-1	Recent	48/48	100	R652-1	Negative	48/48	100	R653-1	Long-term	48/48	100
R651-2	Long-term	48/48	100	R652-2	Long-term	48/48	100	R653-2	Recent	48/48	100
R651-3	Negative	48/48	100	R652-3	Long-term	47/48 **	98	R653-3	Negative	47/48 **	100
R651-4	Recent	47/48 *	98	R652-4	Recent	47/48 **	98	R653-4	Recent	47/48 *	98
A-2023 (Jun 2023)				B-2023 (July 2023)							
R661-1	Long-term	27/27	100	R662-1	Recent	27/27	100				
R661-2	Negative	27/27	100	R662-2	Long-term	26/27 **	96				
R661-3	Recent	27/27	100	R662-3	Negative	27/27	100				

* Misinterpretation ** An error in procedure.

## Data Availability

Data is unavailable due to privacy.

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
