# Peer review of "Establishing Quality Assurance for HIV-1 Rapid Test for Recent Infection in Thailand through the Utilization of Dried Tube Specimens"

_diagnostics, 2024, doi:10.3390/diagnostics14121220_

Round 1
Reviewer 1 Report
Comments and Suggestions for Authors
Minor comments:
Figure 1: please explain what is the difference between V line and LT/R line.
Comments on the Quality of English LanguageEnglish is fine.
Author Response
Please see the attachement

Reviewer 2 Report
Comments and Suggestions for Authors
Review Comments
Title: Establishing Quality Assurance for HIV-1 Rapid test for Recent Infection in Thailand through the Utilization of Dried Tube Specimens
Journal: Diagnostics (MDPI)
The authors examined the performance of rapid test for recent infection by AsanteTM HIV-1 rapid Recency®Assay using the panel samples obtained from de-identified blood units that had ben discarded by the blood donors. The study provided the valuable insist on the usefulness of RTRI in resource-limiting settings. However, I would like to provide the comments as below.
1. Though the authors used the deidentified blood samples from blood donors, it is human subject study, the authors should provide detailed ethical statement if it is waived by concerned ethical committee.
2. The authors should explain what the definitions of recent and long-term HIV are and how to identify in this panel samples, i.e, what are the standard reference tests used to identify and confirm recent and long-term HIV before considering using as panel samples for further assessment.
3. The authors should clearly explain the main difference between QC and EQA assessment in this study and how the authors examined QC and EQA in method section.
4. In method section 2.1, the authors also used Serodia HIV1/2 MIX (Fujirebio Inc, Japan), what is the result of it? Moreover, the authors did not include Serodia test in QC/EQA assessment in section 2.2, what is the reason of excluding Serodia test?
5. The authors should describe how many total samples were included in this study though the authors developed and tested multiple rounds of panel samples in different health facilities/laboratories.
6. Regarding line 102-108 in page 3, the authors should clarify the actual meaning of the description to avoid misunderstanding, First, the authors stated HIV DTS EQA panel contained six DTS tubes, later the authors stated 4 DTS tubes in 2022 and 3 DTS tubes in 2023, what is the main difference in this discrepant number of DTS tubes in distribution thought the quality assurance assessment result was shown on laboratories basis not on number of tests?
7. The authors should explain the characteristics of AsanteTM RTRI in differentiating the recent and long-term HIV, it is supposed that the line in lateral flow kit AsanteTM for LT/R negative means recent and positive means long-term HIV. Additionally, there is some discrepant results between figure 1 and table 3 for 6 weeks, in all figures a, b and c, the sample stored at 45°C showed absence or faint negative control line, i.e, invalid test. The samples stored at RT and 37°C in figure b showed negative V line indicating negative, but it is described as positive in table 3. Similarly, the same samples in figure c showed negative LT/R line, interpreting as recent HIV, but it is described as long-term HIV in table 3. Please clarify these discrepancies.
8. The authors performed the quality assurance of AsanteTM as RTRI test in this study which is HIV-1 rapid test. The limitation exists for being undiagnosed HIV-2 infection through this screening test.
Reviewer 3 Report
Comments and Suggestions for Authors
It is important to prevent the transmission of HIV and to improve the testing of recent HIV infections in patients. This study focuses on establishing quality control and assurance for RTRI in Thailand to accomplish this task. The authors used dried tube specimens as a quality control to monitor the RTRI testing in Thailand. The results, which were very promising, suggest dried tube specimens will be useful in resource limited setting to achieve the 2030 objectives set forth by UNAIDS.
The manuscript reads well and is well written. I satisfied with the data presented by the authors and feel the conclusions made my by the authors from the data are sound and appropriate.
Author Response
Please see the attachement
